# Robust Portfolio Optimization

**Huitong Qiu**
Department of Biostatistics
Johns Hopkins University
Baltimore, MD 21205
hqiu7@jhu.edu

**Fang Han**
Department of Biostatistics
Johns Hopkins University
Baltimore, MD 21205
fhan@jhu.edu

**Han Liu**
Department of Operations Research
and Financial Engineering
Princeton University
Princeton, NJ 08544 hanliu@princeton.edu

**Brian Caffo**
Department of Biostatistics
Johns Hopkins University
Baltimore, MD 21205
bcaffo@jhsph.edu

## Abstract

We propose a robust portfolio optimization approach based on quantile statistics. The proposed method is robust to extreme events in asset returns, and accommodates large portfolios under limited historical data. Specifically, we show that the risk of the estimated portfolio converges to the oracle optimal risk with parametric rate under weakly dependent asset returns. The theory does not rely on higher order moment assumptions, thus allowing for heavy-tailed asset returns. Moreover, the rate of convergence quantifies that the size of the portfolio under management is allowed to scale exponentially with the sample size of the historical data. The empirical effectiveness of the proposed method is demonstrated under both synthetic and real stock data. Our work extends existing ones by achieving robustness in high dimensions, and by allowing serial dependence.

## 1 Introduction

Markowitz's mean-variance analysis sets the basis for modern portfolio optimization theory [1]. However, the mean-variance analysis has been criticized for being sensitive to estimation errors in the mean and covariance matrix of the asset returns [2, 3]. Compared to the covariance matrix, the mean of the asset returns is more influential and harder to estimate [4, 5]. Therefore, many studies focus on the global minimum variance (GMV) formulation, which only involves estimating the covariance matrix of the asset returns.

Estimating the covariance matrix of asset returns is challenging due to the high dimensionality and heavy-tailedness of asset return data. Specifically, the number of assets under management is usually much larger than the sample size of exploitable historical data. On the other hand, extreme events are typical in financial asset prices, leading to heavy-tailed asset returns.

To overcome the curse of dimensionality, structured covariance matrix estimators are proposed for asset return data. [6] considered estimators based on factor models with observable factors. [7, 8, 9] studied covariance matrix estimators based on latent factor models. [10, 11, 12] proposed to shrink the sample covariance matrix towards highly structured covariance matrices, including the identity matrix, order 1 autoregressive covariance matrices, and one-factor-based covariance matrix estimators. These estimators are commonly based on the sample covariance matrix. (sub)Gaussian tail assumptions are required to guarantee consistency.

For heavy-tailed data, robust estimators of covariance matrices are desired. Classic robust covariance matrix estimators include $M$-estimators, minimum volume ellipsoid (MVE) and minimum covari-

ance determinant (MCD) estimators, $S$-estimators, and estimators based on data outlyingness and depth [13]. These estimators are specifically designed for data with very low dimensions and large sample sizes. For generalizing the robust estimators to high dimensions, [14] proposed the Orthogonalized Gnanadesikan-Kettenring (OGK) estimator, which extends [15]'s estimator by re-estimating the eigenvalues; [16, 17] studied shrinkage estimators based on Tyler's $M$-estimator. However, although OGK is computationally tractable in high dimensions, consistency is only guaranteed under fixed dimension. The shrunken Tylor's $M$-estimator involves iteratively inverting large matrices. Moreover, its consistency is only guaranteed when the dimension is in the same order as the sample size. The aforementioned robust estimators are analyzed under independent data points. Their performance under time series data is questionable.

In this paper, we build on a quantile-based scatter matrix[1] estimator, and propose a robust portfolio optimization approach. Our contributions are in three aspects. First, we show that the proposed method accommodates high dimensional data by allowing the dimension to scale exponentially with sample size. Secondly, we verify that consistency of the proposed method is achieved without any tail conditions, thus allowing for heavy-tailed asset return data. Thirdly, we consider weakly dependent time series, and demonstrate how the degree of dependence affects the consistency of the proposed method.

## 2 Background

In this section, we introduce the notation system, and provide a review on the gross-exposure constrained portfolio optimization that will be exploited in this paper.

### 2.1 Notation

Let $\mathbf{v} = (v_1, \ldots, v_d)^{\mathsf{T}}$ be a $d$-dimensional real vector, and $\mathbf{M} = [\mathbf{M}_{jk}] \in \mathbb{R}^{d_1 \times d_2}$ be a $d_1 \times d_2$ matrix with $\mathbf{M}_{jk}$ as the $(j, k)$ entry. For $0 < q < \infty$, we define the $\ell_q$ vector norm of $\mathbf{v}$ as $\|\mathbf{v}\|_q := (\sum_{j=1}^{d} |v_j|)^{1/q}$ and the $\ell_\infty$ vector norm of $\mathbf{v}$ as $\|\mathbf{v}\|_\infty := \max_{j=1}^{d} |v_j|$. Let the matrix $\ell_{\max}$ norm of $\mathbf{M}$ be $\|\mathbf{M}\|_{\max} := \max_{jk} |M_{jk}|$, and the Frobenius norm be $\|\mathbf{M}\|_F := \sqrt{\sum_{jk} M_{jk}^2}$.

Let $\boldsymbol{X} = (X_1, \ldots, X_d)^{\mathsf{T}}$ and $\boldsymbol{Y} = (Y_1, \ldots, Y_d)^{\mathsf{T}}$ be two random vectors. We write $\boldsymbol{X} \overset{\mathrm{d}}{=} \boldsymbol{Y}$ if $\boldsymbol{X}$ and $\boldsymbol{Y}$ are identically distributed. We use $\mathbf{1}, \mathbf{2}, \ldots$ to denote vectors with $1, 2, \ldots$ at every entry.

### 2.2 Gross-exposure Constrained GMV Formulation

Under the GMV formulation, [18] found that imposing a no-short-sale constraint improves portfolio efficiency. [19] relaxed the no-short-sale constraint by a gross-exposure constraint, and showed that portfolio efficiency can be further improved.

Let $\boldsymbol{X} \in \mathbb{R}^d$ be a random vector of asset returns. A portfolio is characterized by a vector of investment allocations, $\mathbf{w} = (w_1, \ldots, w_d)^{\mathsf{T}}$, among the $d$ assets. The gross-exposure constrained GMV portfolio optimization can be formulated as

$$\min_{\mathbf{w}} \mathbf{w}^{\mathsf{T}} \boldsymbol{\Sigma} \mathbf{w} \ \text{ s.t. } \ \mathbf{1}^{\mathsf{T}} \mathbf{w} = 1, \ \|\mathbf{w}\|_1 \leq c. \tag{2.1}$$

Here $\mathbf{1}^{\mathsf{T}} \mathbf{w} = 1$ is the budget constraint, and $\|\mathbf{w}\|_1 \leq c$ is the gross-exposure constraint. $c \geq 1$ is called the gross exposure constant, which controls the percentage of long and short positions allowed in the portfolio [19]. The optimization problem (2.1) can be converted into a quadratic programming problem, and solved by standard software [19].

## 3 Method

In this section, we introduce the quantile-based portfolio optimization approach. Let $Z \in \mathbb{R}$ be a random variable with distribution function $F$, and $\{z_t\}_{t=1}^T$ be a sequence of observations from $Z$. For a constant $q \in [0, 1]$, we define the $q$-quantiles of $Z$ and $\{z_t\}_{t=1}^T$ to be

$$Q(Z; q) = Q(F; q) := \inf\{z : \mathbb{P}(Z \leq z) \geq q\},$$

$$\widehat{Q}(\{z_t\}_{t=1}^T; q) := z^{(k)} \ \text{ where } \ k = \min\Big\{t : \frac{t}{T} \geq q\Big\}.$$

Here $z^{(1)} \leq \ldots \leq z^{(T)}$ are the order statistics of $\{z_t\}_{t=1}^T$. We say $Q(Z; q)$ is unique if there exists a unique $z$ such that $\mathbb{P}(Z \leq z) = q$. We say $\widehat{Q}(\{z_t\}_{t=1}^T; q)$ is unique if there exists a unique $z \in \{z_1, \ldots, z_T\}$ such that $z = z^{(k)}$. Following the estimator $Q_n$ [20], we define the population and sample quantile-based scales to be

$$\sigma^{\mathrm{Q}}(Z) := Q(|Z - \widetilde{Z}|; 1/4) \text{ and } \widehat{\sigma}^{\mathrm{Q}}(\{z_t\}_{t=1}^T) := \widehat{Q}(\{|z_s - z_t|\}_{1 \leq s < t \leq T}; 1/4). \quad (3.1)$$

Here $\widetilde{Z}$ is an independent copy of $Z$. Based on $\sigma^{\mathrm{Q}}$ and $\widehat{\sigma}^{\mathrm{Q}}$, we can further define robust scatter matrices for asset returns. In detail, let $\boldsymbol{X} = (X_1, \ldots, X_d)^{\mathsf{T}} \in \mathbb{R}^d$ be a random vector representing the returns of $d$ assets, and $\{\boldsymbol{X}_t\}_{t=1}^T$ be a sequence of observations from $\boldsymbol{X}$, where $\boldsymbol{X}_t = (X_{t1}, \ldots, X_{td})^{\mathsf{T}}$. We define the population and sample quantile-based scatter matrices (QNE) to be

$$\mathbf{R}^{\mathrm{Q}} := [\mathbf{R}_{jk}^{\mathrm{Q}}] \text{ and } \widehat{\mathbf{R}}^{\mathrm{Q}} := [\widehat{\mathbf{R}}_{jk}^{\mathrm{Q}}],$$

where the entries of $\mathbf{R}^{\mathrm{Q}}$ and $\widehat{\mathbf{R}}^{\mathrm{Q}}$ are given by

$$\mathbf{R}_{jj}^{\mathrm{Q}} := \sigma^{\mathrm{Q}}(X_j)^2, \quad \widehat{\mathbf{R}}_{jj}^{\mathrm{Q}} := \widehat{\sigma}^{\mathrm{Q}}(\{X_{tj}\}_{t=1}^T)^2,$$

$$\mathbf{R}_{jk}^{\mathrm{Q}} := \frac{1}{4}\Big[\sigma^{\mathrm{Q}}(X_j + X_k)^2 - \sigma^{\mathrm{Q}}(X_j - X_k)^2\Big],$$

$$\widehat{\mathbf{R}}_{jk}^{\mathrm{Q}} := \frac{1}{4}\Big[\widehat{\sigma}^{\mathrm{Q}}(\{X_{tj} + X_{tk}\}_{t=1}^T)^2 - \sigma^{\mathrm{Q}}(\{X_{tj} - X_{tk}\}_{t=1}^T)^2\Big].$$

Since $\widehat{\sigma}^{\mathrm{Q}}$ can be computed using $O(T \log T)$ time [20], the computational complexity of $\widehat{\mathbf{R}}^{\mathrm{Q}}$ is $O(d^2 T \log T)$. Since $T \ll d$ in practice, $\widehat{\mathbf{R}}^{\mathrm{Q}}$ can be computed almost as efficiently as the sample covariance matrix, which has $O(d^2 T)$ complexity.

Let $\mathbf{w} = (w_1, \ldots, w_d)^{\mathsf{T}}$ be the vector of investment allocations among the $d$ assets. For a matrix $\mathbf{M}$, we define a risk function $R : \mathbb{R}^d \times \mathbb{R}^{d \times d} \to \mathbb{R}$ by

$$R(\mathbf{w}; \mathbf{M}) := \mathbf{w}^{\mathsf{T}} \mathbf{M} \mathbf{w}.$$

When $\boldsymbol{X}$ has covariance matrix $\boldsymbol{\Sigma}$, $R(\mathbf{w}; \boldsymbol{\Sigma}) = \mathrm{Var}(\mathbf{w}^{\mathsf{T}} \boldsymbol{X})$ is the variance of the portfolio return, $\mathbf{w}^{\mathsf{T}} \boldsymbol{X}$, and is employed as the objected function in the GMV formulation. However, estimating $\boldsymbol{\Sigma}$ is difficult due to the heavy tails of asset returns. In this paper, we adopt $R(\mathbf{w}; \mathbf{R}^{\mathrm{Q}})$ as a robust alternative to the moment-based risk metric, $R(\mathbf{w}; \boldsymbol{\Sigma})$, and consider the following oracle portfolio optimization problem:

$$\mathbf{w}^{\mathrm{opt}} = \operatorname*{argmin}_{\mathbf{w}} R(\mathbf{w}; \mathbf{R}^{\mathrm{Q}}) \text{ s.t. } \mathbf{1}^{\mathsf{T}} \mathbf{w} = 1, \|\mathbf{w}\|_1 \leq c. \quad (3.2)$$

Here $\|\mathbf{w}\|_1 \leq c$ is the gross-exposure constraint introduced in Section 2.2. In practice, $\mathbf{R}^{\mathrm{Q}}$ is unknown and has to be estimated. For convexity of the risk function, we project $\widehat{\mathbf{R}}^{\mathrm{Q}}$ onto the cone of positive definite matrices:

$$\widetilde{\mathbf{R}}^{\mathrm{Q}} = \operatorname{argmin}_{\mathbf{R}} \big\|\widehat{\mathbf{R}}^{\mathrm{Q}} - \mathbf{R}\big\|_{\max}$$
$$\text{s.t. } \mathbf{R} \in S_\lambda := \{\mathbf{M} \in \mathbb{R}^{d \times d} : \mathbf{M}^{\mathsf{T}} = \mathbf{M}, \lambda_{\min} \mathbf{I}_d \preceq \mathbf{M} \preceq \lambda_{\max} \mathbf{I}_d\}. \quad (3.3)$$

Here $\lambda_{\min}$ and $\lambda_{\max}$ set the lower and upper bounds for the eigenvalues of $\widetilde{\mathbf{R}}^{\mathrm{Q}}$. The optimization problem (3.3) can be solved by a projection and contraction algorithm [21]. We summarize the algorithm in the supplementary material. Using $\widetilde{\mathbf{R}}^{\mathrm{Q}}$, we formulate the empirical robust portfolio optimization by

$$\widetilde{\mathbf{w}}^{\mathrm{opt}} = \operatorname*{argmin}_{\mathbf{w}} R(\mathbf{w}; \widetilde{\mathbf{R}}^{\mathrm{Q}}) \text{ s.t. } \mathbf{1}^{\mathsf{T}} \mathbf{w} = 1, \|\mathbf{w}\|_1 \leq c. \quad (3.4)$$

**Remark 3.1.** The robust portfolio optimization approach involves three parameters: $\lambda_{\min}$, $\lambda_{\max}$, and $c$. Empirically, setting $\lambda_{\min} = 0.005$ and $\lambda_{\max} = \infty$ proves to work well. $c$ is typically provided by investors for controlling the percentages of short positions. When a data-driven choice is desired, we refer to [19] for a cross-validation-based approach.

**Remark 3.2.** The rationale behind the positive definite projection (3.3) lies in two aspects. First, in order that the portfolio optimization is convex and well conditioned, a positive definite matrix with lower bounded eigenvalues is needed. This is guaranteed by setting $\lambda_{\min} > 0$. Secondly, the projection (3.3) is more robust compared to the OGK estimate [14]. OGK induces positive definiteness by re-estimating the eigenvalues using the variances of the principal components. Robustness is lost when the data, possibly containing outliers, are projected onto the principal directions for estimating the principal components.

**Remark 3.3.** We adopt the $1/4$ quantile in the definitions of $\sigma^Q$ and $\widehat{\sigma}^Q$ to achieve $50\%$ breakdown point. However, we note that our methodology and theory carries through if $1/4$ is replaced by any absolute constant $q \in (0, 1)$.

# 4 Theoretical Properties

In this section, we provide theoretical analysis of the proposed portfolio optimization approach. For an optimized portfolio, $\widehat{\mathbf{w}}^{\text{opt}}$, based on an estimate, $\mathbf{R}$, of $\mathbf{R}^Q$, the next lemma shows that the error between the risks $R(\widehat{\mathbf{w}}^{\text{opt}}; \mathbf{R}^Q)$ and $R(\mathbf{w}^{\text{opt}}; \mathbf{R}^Q)$ is essentially related to the estimation error in $\mathbf{R}$.

**Lemma 4.1.** *Let $\widehat{\mathbf{w}}^{\text{opt}}$ be the solution to*

$$\min_{\mathbf{w}} R(\mathbf{w}; \mathbf{R}) \ s.t. \ \mathbf{1}^\mathsf{T} \mathbf{w} = 1, \ \|\mathbf{w}\|_1 \leq c \tag{4.1}$$

*for an arbitrary matrix $\mathbf{R}$. Then, we have*

$$|R(\widehat{\mathbf{w}}^{\text{opt}}; \mathbf{R}^Q) - R(\mathbf{w}^{\text{opt}}; \mathbf{R}^Q)| \leq 2c^2 \|\mathbf{R} - \mathbf{R}^Q\|_{\max},$$

*where $\mathbf{w}^{\text{opt}}$ is the solution to the oracle portfolio optimization problem* (3.2)*, and $c$ is the gross-exposure constant.*

Next, we derive the rate of convergence for $R(\widetilde{\mathbf{w}}^{\text{opt}}; \mathbf{R}^Q)$, which relates to the rate of convergence in $\|\widetilde{\mathbf{R}}^Q - \mathbf{R}^Q\|_{\max}$. To this end, we first introduce a dependence condition on the asset return series.

**Definition 4.2.** *Let $\{X_t\}_{t \in Z}$ be a stationary process. Denote by $\mathcal{F}^0_{-\infty} := \sigma(X_t : t \leq 0)$ and $\mathcal{F}^\infty_n := \sigma(X_t : t \geq n)$ the $\sigma$-fileds generated by $\{X_t\}_{t \leq 0}$ and $\{X_t\}_{t \geq n}$, respectively. The $\phi$-mixing coefficient is defined by*

$$\phi(n) := \sup_{B \in \mathcal{F}^0_{-\infty}, A \in \mathcal{F}^\infty_n, \mathbb{P}(B) > 0} |\mathbb{P}(A \mid B) - \mathbb{P}(A)|.$$

*The process $\{X_t\}_{t \in \mathbb{Z}}$ is $\phi$-mixing if and only if $\lim_{n \to \infty} \phi(n) = 0$.*

**Condition 1.** *$\{\boldsymbol{X}_t \in \mathbb{R}^d\}_{t \in \mathbb{Z}}$ is a stationary process such that for any $j \neq k \in \{1, \ldots, d\}$, $\{X_{tj}\}_{t \in \mathbb{Z}}$, $\{X_{tj} + X_{tk}\}_{t \in \mathbb{Z}}$, and $\{X_{tj} - X_{tk}\}_{t \in \mathbb{Z}}$ are $\phi$-mixing processes satisfying $\phi(n) \leq 1/n^{1+\epsilon}$ for any $n > 0$ and some constant $\epsilon > 0$.*

The parameter $\epsilon$ determines the rate of decay in $\phi(n)$, and characterizes the degree of dependence in $\{\boldsymbol{X}_t\}_{t \in \mathbb{Z}}$. Next, we introduce an identifiability condition on the distribution function of the asset returns.

**Condition 2.** *Let $\widetilde{\boldsymbol{X}} = (\widetilde{X}_1, \ldots, \widetilde{X}_d)^\mathsf{T}$ be an independent copy of $\boldsymbol{X}_1$. For any $j \neq k \in \{1, \ldots, d\}$, let $F_{1;j}$, $F_{2;j,k}$, and $F_{3;j,k}$ be the distribution functions of $|X_{1j} - \widetilde{X}_j|$, $|X_{1j} + X_{1k} - \widetilde{X}_j - \widetilde{X}_k|$, and $|X_{1j} - X_{1k} - \widetilde{X}_j + \widetilde{X}_k|$. We assume there exist constants $\kappa > 0$ and $\eta > 0$ such that*

$$\inf_{|y - Q(F; 1/4)| \leq \kappa} \frac{d}{dy} F(y) \geq \eta$$

*for any $F \in \{F_{1;j}, F_{2;j,k}, F_{3;j,k} : j \neq k = 1, \ldots, d\}$.*

Condition 2 guarantees the identifiability of the $1/4$ quantiles, and is standard in the literature on quantile statistics [22, 23]. Based on Conditions 1 and 2, we can present the rates of convergence for $\widehat{\mathbf{R}}^Q$ and $\widetilde{\mathbf{R}}^Q$.

**Theorem 4.3.** *Let $\{\boldsymbol{X}_t\}_{t \in \mathbb{Z}}$ be an absolutely continuous stationary process satisfying Conditions 1 and 2. Suppose $\log d/T \to 0$ as $T \to \infty$. Then, for any $\alpha \in (0, 1)$ and $T$ large enough , with probability no smaller than $1 - 8\alpha^2$, we have*

$$\|\widehat{\mathbf{R}}^Q - \mathbf{R}^Q\|_{\max} \leq r_T. \tag{4.2}$$

*Here the rate of convergence $r_T$ is defined by*

$$r_T = \max\Bigg\{ \frac{2}{\eta^2} \Bigg[ \sqrt{\frac{4(1 + 2C_\epsilon)(\log d - \log \alpha)}{T}} + \frac{4C_\epsilon}{T} \Bigg]^2,$$

$$\frac{4\sigma^Q_{\max}}{\eta} \Bigg[ \sqrt{\frac{4(1 + 2C_\epsilon)(\log d - \log \alpha)}{T}} + \frac{4C_\epsilon}{T} \Bigg] \Bigg\}, \tag{4.3}$$

*where $\sigma^Q_{\max} := \max\{\sigma^Q(X_j), \sigma^Q(X_j + X_k), \sigma^Q(X_j - X_k) : j \neq k \in \{1, \ldots, d\}\}$ and $C_\epsilon := \sum_{k=1}^\infty 1/k^{1+\epsilon}$. Moreover, if $\mathbf{R}^Q \in S_\lambda$ for $S_\lambda$ defined in (3.3), we further have*

$$\|\widetilde{\mathbf{R}}^Q - \mathbf{R}^Q\|_{\max} \leq 2r_T. \tag{4.4}$$

The implications of Theorem 4.3 are as follows.

1. When the parameters $\eta$, $\epsilon$, and $\sigma_{\max}^Q$ do not scale with $T$, the rate of convergence reduces to $O_P(\sqrt{\log d/T})$. Thus, the number of assets under management is allowed to scale exponentially with sample size $T$. Compared to similar rates of convergence obtained for sample-covariance-based estimators [24, 25, 9], we do not require any moment or tail conditions, thus accommodating heavy-tailed asset return data.
2. The effect of serial dependence on the rate of convergence is characterized by $C_\epsilon$. Specifically, as $\epsilon$ approaches 0, $C_\epsilon = \sum_{k=1}^{\infty} 1/k^{1+\epsilon}$ increases towards infinity, inflating $r_T$. $\epsilon$ is allowed to scale with $T$ such that $C_\epsilon = o(T/\log d)$.
3. The rate of convergence $r_T$ is inversely related to the lower bound, $\eta$, on the marginal density functions around the $1/4$ quantiles. This is because when $\eta$ is small, the distribution functions are flat around the $1/4$ quantiles, making the population quantiles harder to estimate.

Combining Lemma 4.1 and Theorem 4.3, we obtain the rate of convergence for $R(\widetilde{\mathbf{w}}^{\mathrm{opt}}; \mathbf{R}^Q)$.

**Theorem 4.4.** *Let $\{\boldsymbol{X}_t\}_{t\in\mathbb{Z}}$ be an absolutely continuous stationary process satisfying Conditions 1 and 2. Suppose that $\log d/T \to 0$ as $T \to \infty$ and $\mathbf{R}^Q \in S_\lambda$. Then, for any $\alpha \in (0,1)$ and $T$ large enough, we have*

$$|R(\widetilde{\mathbf{w}}^{\mathrm{opt}}; \mathbf{R}^Q) - R(\mathbf{w}^{\mathrm{opt}}; \mathbf{R}^Q)| \leq 2c^2 r_T, \tag{4.5}$$

*where $r_T$ is defined in* (4.3) *and $c$ is the gross-exposure constant.*

Theorem 4.4 shows that the risk of the estimated portfolio converges to the oracle optimal risk with parametric rate $r_T$. The number of assets, $d$, is allowed to scale exponentially with sample size $T$. Moreover, the rate of convergence does not rely on any tail conditions on the distribution of the asset returns.

For the rest of this section, we build the connection between the proposed robust portfolio optimization and its moment-based counterpart. Specifically, we show that they are consistent under the elliptical model.

**Definition 4.5.** [26] *A random vector $\boldsymbol{X} \in \mathbb{R}^d$ follows an elliptical distribution with location $\boldsymbol{\mu} \in \mathbb{R}^d$ and scatter $\mathbf{S} \in \mathbb{R}^{d\times d}$ if and only if there exist a nonnegative random variable $\xi \in \mathbb{R}$, a matrix $\mathbf{A} \in \mathbb{R}^{d\times r}$ with $\mathrm{rank}(\mathbf{A}) = r$, a random vector $\boldsymbol{U} \in \mathbb{R}^r$ independent from $\xi$ and uniformly distributed on the $r$-dimensional sphere, $\mathbb{S}^{r-1}$, such that*

$$\boldsymbol{X} \stackrel{\mathrm{d}}{=} \boldsymbol{\mu} + \xi \mathbf{A} \boldsymbol{U}.$$

*Here $\mathbf{S} = \mathbf{A}\mathbf{A}^{\mathsf{T}}$ has rank $r$. We denote $\boldsymbol{X} \sim \mathrm{EC}_d(\boldsymbol{\mu}, \mathbf{S}, \xi)$. $\xi$ is called the generating variate.*

Commonly used elliptical distributions include Gaussian distribution and $t$-distribution. Elliptical distributions have been widely used for modeling financial return data, since they naturally capture many stylized properties including heavy tails and tail dependence [27, 28, 29, 30, 31, 32]. The next theorem relates $\mathbf{R}^Q$ and $R(\mathbf{w}; \mathbf{R}^Q)$ to their moment-based counterparts, $\boldsymbol{\Sigma}$ and $R(\mathbf{w}; \boldsymbol{\Sigma})$, under the elliptical model.

**Theorem 4.6.** *Let $\boldsymbol{X} = (X_1, \ldots, X_d)^{\mathsf{T}} \sim \mathrm{EC}_d(\mu, \mathbf{S}, \xi)$ be an absolutely continuous elliptical random vector and $\widetilde{\boldsymbol{X}} = (\widetilde{X}_1, \ldots, \widetilde{X}_d)^{\mathsf{T}}$ be an independent copy of $\boldsymbol{X}$. Then, we have*

$$\mathbf{R}^Q = m^Q \mathbf{S} \tag{4.6}$$

*for some constant $m^Q$ only depending on the distribution of $\boldsymbol{X}$. Moreover, if $0 < \mathbb{E}\xi^2 < \infty$, we have*

$$\mathbf{R}^Q = c^Q \boldsymbol{\Sigma} \quad \text{and} \quad R(\mathbf{w}; \mathbf{R}^Q) = c^Q R(\mathbf{w}; \boldsymbol{\Sigma}), \tag{4.7}$$

*where $\boldsymbol{\Sigma} = \mathrm{Cov}(\boldsymbol{X})$ is the covariance matrix of $\boldsymbol{X}$, and $c^Q$ is a constant given by*

$$c^Q = Q\left\{\frac{(X_j - \widetilde{X}_j)^2}{\mathrm{Var}(X_j)}; \frac{1}{4}\right\} = Q\left\{\frac{(X_j + X_k - \widetilde{X}_j - \widetilde{X}_k)^2}{\mathrm{Var}(X_j + X_k)}; \frac{1}{4}\right\}$$

$$= Q\left\{\frac{(X_j - X_k - \widetilde{X}_j + \widetilde{X}_k)^2}{\mathrm{Var}(X_j - X_k)}; \frac{1}{4}\right\}. \tag{4.8}$$

*Here the last two inequalities hold when $\mathrm{Var}(X_j + X_k) > 0$ and $\mathrm{Var}(X_j - X_k) > 0$.*

By Theorem 4.6, under the elliptical model, minimizing the robust risk metric, $R(\mathbf{w}; \mathbf{R}^{Q})$, is equivalent with minimizing the standard moment-based risk metric, $R(\mathbf{w}; \mathbf{\Sigma})$. Thus, the robust portfolio optimization (3.2) is equivalent to its moment-based counterpart (2.1) in the population level. Plugging (4.7) into (4.5) leads to the following theorem.

**Theorem 4.7.** *Let $\{\mathbf{X}_t\}_{t\in\mathbb{Z}}$ be an absolutely continuous stationary process satisfying Conditions 1 and 2. Suppose that $\mathbf{X}_1 \sim \mathrm{EC}_d(\boldsymbol{\mu}, \mathbf{S}, \xi)$ follows an elliptical distribution with covariance matrix $\mathbf{\Sigma}$, and $\log d/T \to 0$ as $T \to \infty$. Then, we have*

$$|R(\widetilde{\mathbf{w}}^{\mathrm{opt}}; \mathbf{\Sigma}) - R(\mathbf{w}^{\mathrm{opt}}; \mathbf{\Sigma})| \leq \frac{2c^2}{c^{Q}} r_T,$$

*where $c$ is the gross-exposure constant, $c^{Q}$ is defined in (4.8), and $r_T$ is defined in (4.3).*

Thus, under the elliptical model, the optimal portfolio, $\widetilde{\mathbf{w}}^{\mathrm{opt}}$, obtained from the robust portfolio optimization also leads to parametric rate of convergence for the standard moment-based risk.

## 5 Experiments

In this section, we investigate the empirical performance of the proposed portfolio optimization approach. In Section 5.1, we demonstrate the robustness of the proposed approach using synthetic heavy-tailed data. In Section 5.2, we simulate portfolio management using the Standard & Poor's 500 (S&P 500) stock index data.

The proposed portfolio optimization approach (QNE) is compared with three competitors. These competitors are constructed by replacing the covariance matrix $\mathbf{\Sigma}$ in (2.1) by commonly used covariance/scatter matrix estimators:

1. OGK: The orthogonalized Gnanadesikan-Kettenring estimator constructs a pilot scatter matrix estimate using a robust $\tau$-estimator of scale, then re-estimates the eigenvalues using the variances of the principal components [14].

2. Factor: The principal factor estimator iteratively solves for the specific variances and the factor loadings [33].

3. Shrink: The shrinkage estimator shrinkages the sample covariance matrix towards a one-factor covariance estimator[10].

### 5.1 Synthetic Data

Following [19], we construct the covariance matrix of the asset returns using a three-factor model:
$$X_j = b_{j1}f_1 + b_{j2}f_2 + b_{j3}f_3 + \varepsilon_j, \; j = 1, \ldots, d, \tag{5.1}$$
where $X_j$ is the return of the $j$-th stock, $b_{jk}$ is the loadings of the $j$-th stock on factor $f_k$, and $\varepsilon_j$ is the idiosyncratic noise independent of the three factors. Under this model, the covariance matrix of the stock returns is given by
$$\mathbf{\Sigma} = \mathbf{B}\mathbf{\Sigma}_f\mathbf{B}^{\mathsf{T}} + \mathrm{diag}(\sigma_1^2, \ldots, \sigma_d^2), \tag{5.2}$$
where $\mathbf{B} = [b_{jk}]$ is a $d \times 3$ matrix consisting of the factor loadings, $\mathbf{\Sigma}_f$ is the covariance matrix of the three factors, and $\sigma_j^2$ is the variance of the noise $\varepsilon_i$. We adopt the covariance in (5.2) in our simulations. Following [19], we generate the factor loadings $\mathbf{B}$ from a trivariate normal distribution, $N_d(\boldsymbol{\mu}_b, \mathbf{\Sigma}_b)$, where the mean, $\boldsymbol{\mu}_b$, and covariance, $\mathbf{\Sigma}_b$, are specified in Table 1. After the factor loadings are generated, they are fixed as parameters throughout the simulations. The covariance matrix, $\mathbf{\Sigma}_f$, of the three factors is also given in Table 1. The standard deviations, $\sigma_1, \ldots, \sigma_d$, of the idiosyncratic noises are generated independently from a truncated gamma distribution with shape 3.3586 and scale 0.1876, restricting the support to $[0.195, \infty)$. Again these standard deviations are fixed as parameters once they are generated. According to [19], these parameters are obtained by fitting the three-factor model, (5.1), using three-year daily return data of 30 Industry Portfolios from May 1, 2002 to Aug. 29, 2005. The covariance matrix, $\mathbf{\Sigma}$, is fixed throughout the simulations. Since we are only interested in risk optimization, we set the mean of the asset returns to be $\boldsymbol{\mu} = \mathbf{0}$. The dimension of the stocks under consideration is fixed at $d = 100$.

Given the covariance matrix $\mathbf{\Sigma}$, we generate the asset return data from the following three distributions.

$D_1$: multivariate Gaussian distribution, $N_d(\mathbf{0}, \mathbf{\Sigma})$;

Table 1: Parameters for generating the covariance matrix in Equation (5.2).

| | Parameters for factor loadings | | | Parameters for factor returns | | |
|---|---|---|---|---|---|---|
| $\boldsymbol{\mu}_b$ | $\boldsymbol{\Sigma}_b$ | | | $\boldsymbol{\Sigma}_f$ | | |
| 0.7828 | 0.02915 | 0.02387 | 0.01018 | 1.2507 | -0.035 | -0.2042 |
| 0.5180 | 0.02387 | 0.05395 | -0.00697 | -0.0350 | 0.3156 | -0.0023 |
| 0.4100 | 0.01018 | -0.00697 | 0.08686 | -0.2042 | -0.0023 | 0.1930 |

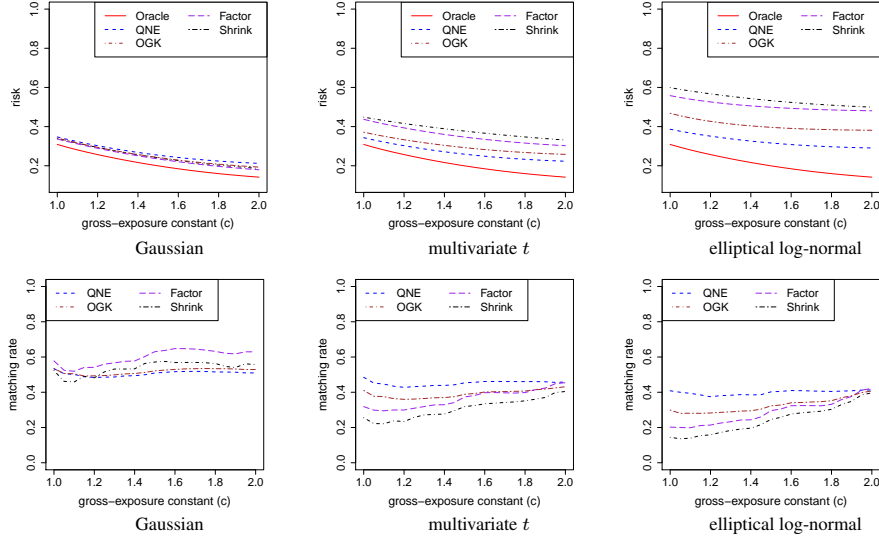

Figure 1: Portfolio risks, selected number of stocks, and matching rates to the oracle optimal portfolios.

$D_2$: multivariate $t$ distribution with degree of freedom 3 and covariance matrix $\boldsymbol{\Sigma}$;

$D_2$: elliptical distribution with log-normal generating variate, $\log N(0,2)$, and covariance matrix $\boldsymbol{\Sigma}$.

Under each distribution, we generate asset return series of half a year ($T = 126$). We estimate the covariance/scatter matrices using QNE and the three competitors, and plug them into (2.1) to optimize the portfolio allocations. We also solve (2.1) with the true covariance matrix, $\boldsymbol{\Sigma}$, to obtain the oracle optimal portfolios as benchmarks. We range the gross-exposure constraint, $c$, from 1 to 2. The results are based on 1,000 simulations.

Figure 1 shows the portfolio risks $R(\widehat{\mathbf{w}}; \boldsymbol{\Sigma})$ and the matching rates between the optimized portfolios and the oracle optimal portfolios[2]. Here the matching rate is defined as follows. For two portfolios $P_1$ and $P_2$, let $S_1$ and $S_2$ be the corresponding sets of selected assets, i.e., the assets for which the weights, $w_i$, are non-zero. The matching rate between $P_1$ and $P_2$ is defined as $r(P_1, P_2) = |S_1 \bigcap S_2| / |S_1 \bigcup S_2|$, where $|S|$ denotes the cardinality of set $S$.

We note two observations from Figure 1. (i) The four estimators leads to comparable portfolio risks under the Gaussian model $D_1$. However, under heavy-tailed distributions $D_2$ and $D_3$, QNE achieves lower portfolio risk. (ii) The matching rates of QNE are stable across the three models, and are higher than the competing methods under heavy-tailed distributions $D_2$ and $D_3$. Thus, we conclude that QNE is robust to heavy tails in both risk minimization and asset selection.

### 5.2 Real Data

In this section, we simulate portfolio management using the S&P 500 stocks. We collect 1,258 adjusted daily closing prices[3] for 435 stocks that stayed in the S&P 500 index from January 1, 2003

Table 2: Annualized Sharpe ratios, returns, and risks under 4 competing approaches, using S&P 500 index data.

|  |  | QNE | OGK | Factor | Shrink |
|---|---|---|---|---|---|
| Sharpe ratio | c=1.0 | **2.04** | 1.64 | 1.29 | 0.92 |
|  | c=1.2 | **1.89** | 1.39 | 1.22 | 0.74 |
|  | c=1.4 | **1.61** | 1.24 | 1.34 | 0.72 |
|  | c=1.6 | **1.56** | 1.31 | 1.38 | 0.75 |
|  | c=1.8 | **1.55** | 1.48 | 1.41 | 0.78 |
|  | c=2.0 | **1.53** | 1.51 | 1.43 | 0.83 |
| return (in %) | c=1.0 | **20.46** | 16.59 | 13.18 | 9.84 |
|  | c=1.2 | **18.41** | 13.15 | 10.79 | 7.20 |
|  | c=1.4 | **15.58** | 11.30 | 10.88 | 6.55 |
|  | c=1.6 | **15.02** | 11.48 | 10.68 | 6.49 |
|  | c=1.8 | **14.77** | 12.39 | 10.57 | 6.58 |
|  | c=2.0 | **14.51** | 12.27 | 10.60 | 6.76 |
| risk (in %) | c=1.0 | **10.02** | 10.09 | 10.19 | 10.70 |
|  | c=1.2 | 9.74 | 9.46 | **8.83** | 9.76 |
|  | c=1.4 | 9.70 | 9.10 | **8.12** | 9.14 |
|  | c=1.6 | 9.63 | 8.75 | **7.71** | 8.68 |
|  | c=1.8 | 9.54 | 8.39 | **7.51** | 8.38 |
|  | c=2.0 | 9.48 | 8.13 | **7.43** | 8.18 |

to December 31, 2007. Using the closing prices, we obtain 1,257 daily returns as the daily growth rates of the prices.

We manage a portfolio consisting of the 435 stocks from January 1, 2003 to December 31, 2007[4]. On days $i = 42, 43, \ldots, 1, 256$, we optimize the portfolio allocations using the past 2 months stock return data (42 sample points). We hold the portfolio for one day, and evaluate the portfolio return on day $i + 1$. In this way, we obtain 1,215 portfolio returns. We repeat the process for each of the four methods under comparison, and range the gross-exposure constant $c$ from 1 to 2[5].

Since the true covariance matrix of the stock returns is unknown, we adopt the Sharpe ratio for evaluating the performances of the portfolios. Table 2 summarizes the annualized Sharpe ratios, mean returns, and empirical risks (i.e., standard deviations of the portfolio returns). We observe that QNE achieves the largest Sharpe ratios under all values of the gross-exposure constant, indicating the lowest risks under the same returns (or equivalently, the highest returns under the same risk).

## 6 Discussion

In this paper, we propose a robust portfolio optimization framework, building on a quantile-based scatter matrix. We obtain non-asymptotic rates of convergence for the scatter matrix estimators and the risk of the estimated portfolio. The relations of the proposed framework with its moment-based counterpart are well understood.

The main contribution of the robust portfolio optimization approach lies in its robustness to heavy tails in high dimensions. Heavy tails present unique challenges in high dimensions compared to low dimensions. For example, asymptotic theory of $M$-estimators guarantees consistency in the rate $O_P(\sqrt{d/n})$ even for non-Gaussian data [34, 35]. If $d \ll n$, statistical error diminishes rapidly with increasing $n$. However, when $d \gg n$, statistical error may scale rapidly with dimension. Thus, stringent tail conditions, such as subGaussian conditions, are required to guarantee consistency for moment-based estimators in high dimensions [36]. In this paper, based on quantile statistics, we achieve consistency for portfolio risk without assuming any tail conditions, while allowing $d$ to scale nearly exponentially with $n$.

Another contribution of his work lies in the theoretical analysis of how serial dependence may affect consistency of the estimation. We measure the degree of serial dependence using the $\phi$-mixing coefficient, $\phi(n)$. We show that the effect of the serial dependence on the rate of convergence is summarized by the parameter $C_\epsilon$, which characterizes the size of $\sum_{n=1}^{\infty} \phi(n)$.

## Footnotes

[1]A scatter matrix is defined to be any matrix proportional to the covariance matrix by a constant.

[2]Due to the $\ell_1$ regularization in the gross-exposure constraint, the solution is generally sparse.

[3]The adjusted closing prices accounts for all corporate actions including stock splits, dividends, and rights offerings.

[4]We drop the data after 2007 to avoid the financial crisis, when the stock prices are likely to violate the stationary assumption.

[5]$c = 2$ imposes a 50% upper bound on the percentage of short positions. In practice, the percentage of short positions is usually strictly controlled to be much lower.

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
