[Supplementary Material · RPO-supp.pdf]

# Supplement to Robust Portfolio Optimization

**Huitong Qiu**
Department of Biostatistics
Johns Hopkins University
Baltimore, MD 21205
hqiu7@jhu.edu

**Fang Han**
Department of Biostatistics
Johns Hopkins University
Baltimore, MD 21205
fhan@jhu.edu

**Han Liu**
Department of Operations Research
and Financial Engineering
Princeton University
Princeton, NJ 08544 hanliu@princeton.edu

**Brian Caffo**
Department of Biostatistics
Johns Hopkins University
Baltimore, MD 21205
bcaffo@jhsph.edu

## A  Matrix Projection

In this section, we summarize the algorithm proposed in [1] for solving the matrix projection problem (3.3). Let

$$\Omega_1 := \Big\{ \boldsymbol{x} = \text{vec}(\mathbf{X}) : \mathbf{X} \in S_\lambda \Big\}$$

$$\Omega_2 := \Big\{ \boldsymbol{z} = \text{vec}(\mathbf{Z}) : \mathbf{Z} \in \mathbb{R}^{d \times d}, \mathbf{Z} = \mathbf{Z}^\mathsf{T}, \sum_{i,j=1}^{d} |Z_{ij}| \leq 1 \Big\}.$$

For any symmetric matrix $\mathbf{V} \in \mathbb{R}^{d \times d}$ and $\mathbf{v} = \text{vec}(\mathbf{V})$, define the projection of $\mathbf{v}$ onto $\Omega_i$ as

$$P_{\Omega_i}(\mathbf{v}) = \arg \min_{\boldsymbol{x} \in \Omega_i} \Big\| \boldsymbol{x} - \mathbf{v} \Big\|_2^2, \tag{A.1}$$

for $i = 1, 2$. The algorithm for solving (3.3) builds on solutions to the problems in (A.1). Solving for $P_{\Omega_1}(\mathbf{v})$ is straightforward. It's well known that

$$P_{\Omega_1}(\mathbf{v}) = \text{vec}(\mathbf{U}\widetilde{\mathbf{\Lambda}}\mathbf{U}^\mathsf{T}), \tag{A.2}$$

where $\mathbf{V} = \mathbf{U}\mathbf{\Lambda}\mathbf{U}^\mathsf{T}$ is a spectral decomposition of $\mathbf{V}$, $\widetilde{\mathbf{\Lambda}} = \text{diag}(\widetilde{\mathbf{\Lambda}}_{11}, \ldots, \widetilde{\mathbf{\Lambda}}_{dd})$ and $\widetilde{\mathbf{\Lambda}}_{ii} = \min\Big\{ \max\{\mathbf{\Lambda}_{ii}, \lambda_{\min}\}, \lambda_{\max} \Big\}$ for $i = 1, \ldots, d$.

Next we solve for $P_{\Omega_2}(\mathbf{v})$. Let $\text{sign}(\mathbf{v}) = \{\text{sign}(v_1), \ldots, \text{sign}(v_d)\}^\mathsf{T}$ be a vector of the signs of $\mathbf{v}$'s entries. Denote $|\mathbf{v}| = \text{sign}(\mathbf{v}) \circ \mathbf{v}$ and $\widetilde{\mathbf{v}} = T_{|\mathbf{v}|}(|\mathbf{v}|)$, where $T_{|\mathbf{v}|}$ is a permutation transformation that sorts the elements of $|\mathbf{v}|$ in descending order. Now, if $\mathbf{1}^\mathsf{T}\widetilde{\mathbf{v}} \leq 1$, we set $(\widetilde{\boldsymbol{x}}, \widetilde{y}) = (\widetilde{\mathbf{v}}, 0)$. If $\mathbf{1}^\mathsf{T}\widetilde{\mathbf{v}} > 1$, let $\Delta\mathbf{v} := (\widetilde{v}_1 - \widetilde{v}_2, \ldots, \widetilde{v}_{d-1} - \widetilde{v}_d, \widetilde{v}_d)^\mathsf{T} \in \mathbb{R}^d$. Note that $\Delta v_i \geq 0$ for $i = 1, \ldots, d$ and $\sum_{i=1}^{d} i \Delta v_i = \mathbf{1}^\mathsf{T}\widetilde{\mathbf{v}} > 1$. Thus, there exists a smallest integer $K$ such that $\sum_{i=1}^{K} i \Delta v_i \geq 1$. In this case, we set

$$\widetilde{y} = \frac{1}{K} \Big( \sum_{i=1}^{K} \widetilde{v}_i - 1 \Big) \text{ and } \widetilde{\boldsymbol{x}} = (\widetilde{v}_1 - \widetilde{y}, \ldots, \widetilde{v}_K - \widetilde{y}, 0, \ldots, 0)^\mathsf{T} \in \mathbb{R}^d.$$

Now we can express $P_{\Omega_2}(\mathbf{v})$ as

$$P_{\Omega_2}(\mathbf{v}) = \text{sign}(\mathbf{v}) \circ T_{|\mathbf{v}|}^{-1}(\widetilde{\boldsymbol{x}}). \tag{A.3}$$

---

**Algorithm 1** Solving matrix projection problem (3.3)

---

$\widetilde{\mathbf{R}}^{\mathrm{Q}} \leftarrow \mathrm{MatrixProjection}(\widehat{\mathbf{R}}^{\mathrm{Q}}, \lambda_{\min}, \lambda_{\max}, \boldsymbol{x}^0, \boldsymbol{z}^0, \gamma, \epsilon, N)$
$\boldsymbol{r} \leftarrow \mathrm{vec}(\widehat{\mathbf{R}}^{\mathrm{Q}})$
**for** $k = 0, \ldots, N$ **do**
&emsp;$\boldsymbol{e}_x^k \leftarrow \boldsymbol{x}^k - P_{\Omega_1}(\boldsymbol{x}^k - \boldsymbol{z}^k)$
&emsp;$\boldsymbol{e}_z^k \leftarrow \boldsymbol{z}^k - P_{\Omega_2}(\boldsymbol{z}^k + \boldsymbol{x}^k - \boldsymbol{r})$
&emsp;$\boldsymbol{e}^k \leftarrow (\boldsymbol{e}_x^k, \boldsymbol{e}_z^k)^{\mathsf{T}}$
&emsp;**if** $\|\boldsymbol{e}^k\|_{\max} < \epsilon$, **then**
&emsp;&emsp;**break**
&emsp;**else**
&emsp;&emsp;$\boldsymbol{x}^{k+1} \leftarrow \boldsymbol{x}^k - \gamma(\boldsymbol{e}_x^k - \boldsymbol{e}_z^k)/2$
&emsp;&emsp;$\boldsymbol{z}^{k+1} \leftarrow \boldsymbol{z}^k - \gamma(\boldsymbol{e}_x^k + \boldsymbol{e}_z^k)/2$
&emsp;**end if**
**end for**
**return** $\widetilde{\mathbf{R}}^{\mathrm{Q}} = \mathrm{mat}(\boldsymbol{x}^k)$

---

Next we solve the matrix projection problem in (3.3). Recall that $\widehat{\mathbf{R}}^{\mathrm{Q}}$ is the matrix to be projected to $S_\lambda$. Since for any vector $\boldsymbol{y} \in \mathbb{R}^d$, we have $\|\boldsymbol{y}\|_{\max} = \max_{\boldsymbol{c} \in \mathbb{R}^d, \|c\|_1 \leq 1} \boldsymbol{c}^{\mathsf{T}} \boldsymbol{y}$, it follows that problem (3.3) can be reformulated as the following mini-max problem:

$$\min_{\boldsymbol{x} \in \Omega_1} \max_{\boldsymbol{z} \in \Omega_2} \boldsymbol{z}^{\mathsf{T}} \left\{ \boldsymbol{x} - \mathrm{vec}\left(\widehat{\mathbf{R}}^{\mathrm{Q}}\right) \right\}. \tag{A.4}$$

If $(\boldsymbol{x}^{\mathrm{opt}}, \boldsymbol{z}^{\mathrm{opt}})$ is a solution to problem (A.4), then $\mathrm{mat}(\boldsymbol{x}^{\mathrm{opt}})$ is a solution to problem (3.3). Algorithm 1 gives the pseudo code for solving problem (A.4), and thus (3.3). Recall that $0 \leq \lambda_{\min} < \lambda_{\max} \leq \infty$ are the lower and upper bounds of the eigenvalues of the projection. $\boldsymbol{x}^0 \in \Omega_1$ and $\boldsymbol{z}^0 \in \Omega_2$ are arbitrary initial points. $\gamma \in (0, 2)$ is a parameter controlling the step lengths of every iteration. $\epsilon > 0$ is a prespecified tolerance level. $N \in \mathbb{N}$ is the maximum number of iterations desired. The convergence of Algorithm 1 is guaranteed by the following theorem.

**Theorem A.1** ([1]). *Let $\boldsymbol{u}^{\mathrm{opt}} := (\boldsymbol{x}^{\mathrm{opt}}, \boldsymbol{z}^{\mathrm{opt}})$ be a solution to (A.4). Denote $\boldsymbol{u}^k := (\boldsymbol{x}^{k^{\mathsf{T}}}, \boldsymbol{z}^{k^{\mathsf{T}}})^{\mathsf{T}}$ and $\boldsymbol{e}_u^k := (\boldsymbol{e}_x^{k^{\mathsf{T}}}, \boldsymbol{e}_z^{k^{\mathsf{T}}})^{\mathsf{T}}$. Then Algorithm 1 produces a sequence $\{\boldsymbol{u}^k\}$ satisfying*

$$\left\|\boldsymbol{u}^{k+1} - \boldsymbol{u}^{\mathrm{opt}}\right\|^2 \leq \left\|\boldsymbol{u}^k - \boldsymbol{u}^{\mathrm{opt}}\right\|^2 + \frac{\gamma(2-\gamma)}{2}\left\|\boldsymbol{e}_u^k\right\|^2.$$

## References

[1] M. H. Xu and H. Shao. Solving the matrix nearness problem in the maximum norm by applying a projection and contraction method. *Advances in Operations Research*, 2012:1–15, 2012.