[Reviews · NeurIPS 2015]

Submitted by Assigned_Reviewer_1

The paper presents a method to derive a portfolio based on minimizing the variance of the return estimator, where the covariance is estimated by a quantile-based scatter matrix. The authors discuss how to estimate this matrix R^Q from data. An important property of their estimator is that no assumptions on the tails of the distribution are made. In Section 4, the authors bound the error of the quantile-based scatter matrix and their empirical estimate. Moreover, they show that R^Q is proportional to the covariance matrix if the asset returns follow an elliptical distribution (including Gaussians).

I'm inclined to accept the paper because especially the latter finding is of interest to the ML community. However, the authors should address the following issues.

Minimizing the overall variance of the financial return estimator in Eq. 2.1 is a "proxy" to solve the actual problem of maximizing return. Even if the problem statement and terms such as long/short positions are well known in the finance community, they may be not in the ML community. Please provide a clear problem setting. In contrast, there is no need to introduce matrix or vector norms.

Replacing the covariance by any matrix R^Q seems arbitrary and unmotivated in the first place. If I understand correctly, the actual problem is that the sample variance is a "bad" estimate of the real covariance (for several reasons including a lack of observations). The proposed estimator R-tilde^Q is supposed to be a more accurate estimate w.r.t. metric E[X^T,w], where w is defined by solving (3.2) given the covariance estimate and X is any stochastic process of asset returns. This should be made clear in the beginning of Section 3. Also I would like to see an empirical comparison of different covariance estimators for various matrix distances; not just for E[X^T,w] as in Figure 1.

Is estimator Q-hat in (3.1) a consistent/unbiased estimator of Q? To me it seems that Q-hat overestimates the quantiles since pairs (z_s,z_t) are "drawn" from sample {z_i} without replacement.

In terms of notation, there is no real distinction between random variables and their realization which is sometimes confusing.

typos: line 70 and 71, M_jk should not be bold face line 72: ^q missing for the q-norm definition 4.2: P(A) - P(A|B) without |.| Eq. 8 contains equalities; the text thereafter refers to inequalities
Summary: The authors derive an estimator of a "proxy" of the covariance matrix of a stationary stochastic process (in their case asset returns) which is robust to data outliers and does not make assumptions on the tails of the distribution. They show that for elliptical distributions, which includes Gaussians, this proxy is consistent with true covariance matrix up to a scaling factor; and that their proposed estimator of the proxy has bounded error. I'm inclined to accept the paper because of this finding; while the there is room for improvement in the presentation of the work.

Submitted by Assigned_Reviewer_2

This paper proposes a robust portfolio optimization approach based on quantile statistics. The paper seems technically sound but authors need to survey related works about portfolio optimization.

For example, the paper, Robust Asset Allocation, http://citeseerx.ist.psu.edu/viewdoc/download?doi=10.1.1.136.3697&rep=rep1&type=pdf proposes a similar problem (Eq.(7) in p.5). The authors need to discuss the relation of the proposed model to related models.

The title "Robust Portfolio Optimization" is also very confusing. There is already "Robust Portfolio Optimization" used in portfolio optimization, which refers to the portfolio optimization whose uncertainty is described with robust optimization technique. Check the book, Robust Portfolio Optimization and Management, http://as.wiley.com/WileyCDA/WileyTitle/productCd-047192122X.html

By searching for the keyword "robust portfolio optimization" or "value-at-risk" online, the authors will find various related works which have similar motivation to the authors' work.
Summary: The paper seems technically sound but authors need to survey related works about portfolio optimization.

Submitted by Assigned_Reviewer_3

The paper proposes an estimator for the covariance or scatter matrix. It is based on quantile statistics followed by a projection which guarantees the eigenvalues are bounded away from zero by a fixed constant. Theoretical analysis is then provided which looks at the regret vs an oracle of the obtained risk (risk meaning portfolio variance obtained under a type of minimum variance optimisation).

The theory looks good, but it's all proven in the supplementary materials which are very long.

The experimental results on synthetic data seem to support the utility of the estimator. I think it would be interesting to do more synthetic data experiments, including those where the proposed method is not the best.

The real data experiment in table 2 are not convincing, in fact they seem to be negative evidence for the utility of the method. To be fair, real world stock data is notoriously hard to model, and the paper is primarily theoretical. Still, it is strange that your method seems to dominate only in terms of mean return. As I understand, none of the methods consider forecasting the mean return (which is reasonable), and moreover your theoretical statements are about the risk. So it was dissuading to see results where the risk is not better for your method, but the mean is. Then, including the Sharpe ratio is, like the inclusion of the mean return, somewhat irrelevant given your setup.
Summary: Proposes a quantile based estimator for the covariance matrix for use in minimum variance portfolio optimisation, and theoretically analyses the difference in risk vs an oracle.

Submitted by Assigned_Reviewer_4

The paper describes and analysies portfolio optimization with a quantile-based statistic that is proportional to the data covariance for an important class of distribtions. The benefit in comparison to using the sample covariance in the optimization problem is a faster convergence for large portfolios and few observation periods.

This reviewer -- who is not an expert in the field of portfolio management nor in quantile statistics -- found this paper interesting to read and a good starting point for further thought. Still, there are many things to improve:

a) Experiments 1: It is claimed throughout that the specific purpose of this paper is the case T much smaller than d. However, in the examples is it T greater than d, twice.

b) Experiments 2: Leaving out the data for year 2007 seems practical. But managing 2007 would have been a nice example that the method can deal with heavy tails.

c) Experiments 3: Why are the returns of the proposed method especially high? I thought the approach only minimzes the variance. This may indicate that the experimental results are rather arbitrary.

d) Writing 1: Apart from some misspellings (see below, line numbers would have been nice!) several non-standard terms are first used and only explained later, e.g. matching rate, breakdown point, \Sigma in (2.1).

e) Writing 2: It would be helpful to decribe the intuition behind the quantile-based scales.

This reviewer could not check the axact maths of the forumlations in the paper nor the proofs of the theorems in the appendix. Similarly, judging the novelty of the work was difficult for this reviewer due to limited experience in the field. Following the arguments in the paper, applying the described statistic to portfolio optimization as well as the extensive analysis seems sufficiently novel.

Spelling:

- ||v||_q: an exponent is missing

- \hat{R}^Q_jk: the hat on the second \sigma is missing

- .. shrinkage estimator shrinks ..

Summary: For this non-expert reviewer, it is an interesting work with extensive theoretical analysis, yet with questionable experimental results.

Submitted by Assigned_Reviewer_5

Provide definitions of: sharp ratio, return and risk

I like that the authors provide simulated and real data and spend some space to explain the complicated way the data are generated. This is necessary because without such an analysis the methods could not be judged.

For the results in table 2, I would like to see SE. Without these I am not sure how to interpret the results. However, even without one can say that the new method leads to good results.

We hold the portfolio for one day, and evaluate the portfolio return on day i + 1. In this way, we obtain 1,215 portfolio returns. How do you come to this number?

Summary: A portfolio optimization is studied and a method based on quantile statistics proposed.

Author Feedback
Author rebuttal: We would like to first thank all referees for their very helpful and constructive comments. We will revise the manuscript following their comments.

Below please find the responses to some specific comments.

Referee #1.
(i) Is estimator Q-hat in (3.1) a consistent/unbiased estimator of Q? To me it seems that Q-hat overestimates the quantiles since pairs (z_s,z_t) are "drawn" from sample {z_i} without replacement.

Ans: Q-hat in (3.1) is a consistent (but not necessarily unbiased) estimator of Q. This is consistent to the theory that "the sample quantile is a biased, but asymptotically unbiased, estimator of the population quantile". We refer to Lemma A.5 for technical details.

Referee #2.
(i) The real data experiment in table 2 are not convincing, in fact they seem to be negative evidence for the utility of the method... Still, it is strange that your method seems to dominate only in terms of mean return... So it was dissuading to see results where the risk is not better for your method, but the mean is...

Ans: We are sorry for the confusion here. In Table 2 we measure the risks using the sample standard deviation (SDV). Actually, we also have the results using the quantile-based risk metric. They confirm the minima of the risks (using this robust risk measurement) are also attained via using our proposed method (which also suggests the stock return data are heavy-tailed). Due to the space limit and for presentation clearness, we did not add these results to the main manuscript. But we will add them back after being aware of this confusion.

Referee #3
(i) Why are the returns of the proposed method especially high? I thought the approach only minimizes the variance. This may indicate that the experimental results are rather arbitrary.

Ans: Please check the response to Referee #2. Again, we feel sorry for this confusion.

Referee #4
(i) We hold the portfolio for one day, and evaluate the portfolio return on day i + 1. In this way, we obtain 1,215 portfolio returns. How do you come to this number?

Ans: We are sorry for the confusion here. We get the number 1,215 by counting the number of trading days from the beginning of March (day i = 42) in 2003 to the end of 2007 (day i = 1,256).

Referee #5

We thank the referee for her/his positive comments.